# Non-Simultaneous Bilateral Ischemic Optic Neuropathy Related to High Altitude and Airplane Flight in a Patient with Cerebral Small Vessel Disease

**DOI:** 10.3390/diagnostics11122325

**Published:** 2021-12-10

**Authors:** Ana Boned-Murillo, Olivia Esteban-Floria, Mireya Martinez-Velez, Javier Mateo Gabas, Francisco Javier Ascaso Puyuelo

**Affiliations:** 1Department of Ophthalmology, Hospital Clínico Universitario Lozano Blesa, CP 50009 Zaragoza, Spain; oliviaestebanfloria@hotmail.com (O.E.-F.); mireyamvelez@gmail.com (M.M.-V.); jmateo_1999@yahoo.com (J.M.G.); jascaso@gmail.com (F.J.A.P.); 2Aragon Health Research Institute (IIS Aragón), CP 50009 Zaragoza, Spain

**Keywords:** cerebral small vessel disease, high altitude, hypoxia, non-arteritic anterior ischemic optic neuropathy (NA-AION)

## Abstract

Non-arteritic anterior ischemic optic neuropathy (NA-AION) is considered the most frequent type of acute optic neuropathy. A 61-year-old woman presented with a NA-AION in her right eye within 24 h following an airplane flight. One year later, after driving for 10 days with a daily accumulated altitude of 1500 m, she developed a NA-AION in her left eye. Systemic disorders were investigated, and cerebral small vessel disease was observed via cranial computed tomography. An inadequate response to hypoxia, in a patient with individual susceptibility, could lead to reduced blood supply to the optic nerve head, which could represent an underlying cause of NA-AION.

## 1. Introduction

Non-arteritic anterior ischemic optic neuropathy (NA-AION) is the most common type of acute optic neuropathy, and is one of the main causes of seriously impaired vision in middle-aged patients, with a reported annual incidence of 2.3–10.2 per 100,000 population over 50 years, with a significantly higher frequency in Caucasians [1]. The lower visual field is usually affected, with a normal visual acuity in 40% of patients; relative afferent pupillary defect (RAPD) is always present, and disc oedema is sometimes observed, associated with flame shaped hemorrhages at the margins [1]. Contrary to the arteritic form (A-AION), known as giant cell arteritis (GCA), it is not necessarily associated with the elderly, constitutional symptoms, elevated levels of erythrocyte sedimentation rate (ESR) or C-reactive protein (CRP), or thrombocytosis. Vision can worsen over the 2-week period following initial presentation, and typically stabilizes in 2 months; visual acuity improvement occurs in less than 10% of patients, as well as visual field improvement, and there is never complete recovery of both acuity and visual field [1]. Rates of recurrence in the affected eye range from 3% to 8%, while fellow eye involvement ranges from 15% to 24%, over 5 years [1]. Although the pathophysiology is not completely understood, some evidence supports the theory that NA-AION occurs following a transient loss of perfusion support—specifically when support drops to below critical autoregulatory levels—from the posterior ciliary artery, which supplies the optic nerve head (ONH) [2], producing an ischemic lesion involving the junction between two adjacent arterial territories. Besides the classic ocular and systemic risk factors, such as arterial hypertension, diabetes mellitus, and dyslipidemia, NA-AION has been associated with exposure to high-altitude conditions [3,4,5,6]. The present case demonstrates a rare, non-simultaneous bilateral acute NA-AION, related to high altitude and airplane flight, in a patient with small vessel disease (SVD). 

## 2. Case Report

A 61-year-old woman was referred to our clinic complaining of an isolated, sudden, and painless visual loss in her right eye, within 24 h following a 2 h airplane flight (at 30,000 feet) from Paris to Madrid. Her medical history showed well-controlled hypercholesterolemia. 

Twenty-four hours later, best-corrected visual acuity (BCVA) was 20/200 in her right eye (RE) and 20/20 in her left eye (LE). There was a relative afferent pupillary defect (RAPD), color vison deficiency, and an inferior hemifield and temporal-superior quadrant scotoma (Figure 1A) in the RE; funduscopic examination revealed a 360° swelling of the right optic disc, with superonasal flame-shaped hemorrhaging, venous congestion, and tortuosity. The LE was normal, with a cup-to-disc ratio of less than 0.1, suggesting “disc-at-risk”. Fundus fluorescein angiography (FFA) of the RE showed late optic-disc staining (Figure 2). Cardiac and carotid Doppler ultrasound, autoimmune, and hypercoagulability tests were normal, with the exception of slightly raised serum cholesterol levels. Cranial computed tomography (CT) revealed previously unknown white matter lesions (Figure 3). NA-AION associated with cerebral SVD was diagnosed. 

After one year of treatment with aspirin (100 mg daily), the patient developed visual disturbances in her LE, occurring during a 10 days drive in the French Alps, with a daily accumulated altitude of 1500 m. BCVA was 20/200 in her RE and 20/40 in her LE. Examination revealed edematous and flame-shaped retinal hemorrhaging at the border of the left ONH, vascular tortuosity, fluorescein leakage (during FFA), and severe widespread visual field loss with central-sparing (Figure 1B), suggesting a NA-AION in the LE. 

At the time of publication, BCVA had decreased to 20/200 in both eyes, fundus examination showed bilateral optic disc atrophy, and spectral domain optical coherence tomography had confirmed a marked decrease in peripapillary retinal nerve fiber layer thickness in both eyes (Figure 4). 

## 3. Discussion

Autoregulation of retinal blood flow plays a role in maintaining normal circulation in eyes suffering from chronic hypoxia, as may occur at high altitude [3]. Mullner et al. [4] proposed that high-altitude retinopathy (HAR) can be observed in climbers with dysfunction in the ocular vascular autoregulatory response. They measured retinal perfusion in mountaineers following an expedition in Himalaya; after the expedition, two of the eight climbers had bilateral retinal hemorrhage. On the other hand, an increase in retinal blood flow, between 18 and 96%, was caused by chronic hypoxic exposure in five climbers. What is more, two climbers suffered a decrease in retinal blood flow of approximately 21–31%. Thus, a dysfunction in the ocular vascular autoregulatory response, under conditions of chronic hypoxia, may lead to spasming of the retinal vessels, and reduce perfusion to the ONH; this condition is a proposed mechanism for the development of NA-AION.

HAR is characterized by ocular signs, including retinal hemorrhaging, dilation and tortuosity of the retinal vessels, and optic disc edema [7,8]. Only two cases of ischemic optic neuropathy related to high altitude have been previously reported: the first reports a unilateral NA-AION in a man who stayed at an altitude of 5472 m above sea level for three months [9]; the second reports a bilateral NA-AION in a 66 year old woman during a trek at altitudes higher than 2500 m [6]. According to the authors, the combination of high altitude and effort induced prolonged desaturation of oxyhemoglobin, causing ischemia of the ONH. Visual loss linked to hypoxia in the ONH has also been associated with long-duration flight: Kaisserman and Frucht-Pery [5] reported a case of AION following a 15 h transatlantic flight, in which a reduction in cabin pressure was linked to decreased oxygen saturation and increased levels of CO_2_ in the blood of an otherwise healthy passenger. Therefore, while internal adaptive mechanisms work to maintain a normal blood perfusion to the ONH under conditions of physiological stress, high attitude conditions can trigger a dysfunction of autoregulatory blood-flow mechanisms, leading to hypoxia, and, in some cases, triggering AION.

Systemic, endocrine, neural factor, and oxidative stress disorders of the neurovascular unit (NVU) can contribute to the development of a brain injury, and play a major role in dysregulation, neuroinflammation, and neurodegeneration after a stroke. Complex communication and cell-to-cell communication is essential [10,11]; for example, pericyte modulators, such as the NADPH oxidase Nox4 (reactive oxygen species (ROS)), are significantly increased under hypoxic conditions, increasing MMP-9 and possibly increasing the size of the infarct area by playing a harmful role in the acute phase of ischemic stroke [11]. 

In our patient, underlying cerebral SVD (a condition that been linked to stroke and vascular dementia) was observed. Cerebral SVD is radiologically visible as a low signal on CT imaging of the brain, often as lacunar infarcts or white matter lesions (as observed in this case), and is associated with problems in the small perforating arteries and arterioles that support deep-brain vasculature [12]. Retinal and cerebral small vessels are developmentally related, of similar size, and share physiologic characteristics [13]. As Moran et al. [12] reported, damage to retinal small vessels is suggestive of arteriosclerosis, and is commonly seen in patients with deep-brain strokes. Association between cerebral SVD and NA-AION was studied by Kim et al. [14], who found an association after adjusting for age, sex, and medical histories. Although NA-AION seems to be linked to SVD affecting vessels that supply the ONH, the exact mechanism is not completely understood; however, as NA-AION is typically observed in patients of middle age or older, and with vascular risk factors for atherosclerosis [12], a role in its etiology is suggested.

In summary, this case report represents an unusual case—and the first reported in literature—of bilateral ischemic optic neuropathy. Although ischemic disorders of the optic nerve are considered to have a complex etiology, we propose that an inadequate response to hypoxia in a patient with individual susceptibility (e.g., associated with cerebral SVD) could lead to reduced blood supply to the ONH and trigger AION. 

## Figures and Tables

**Figure 1 diagnostics-11-02325-f001:**
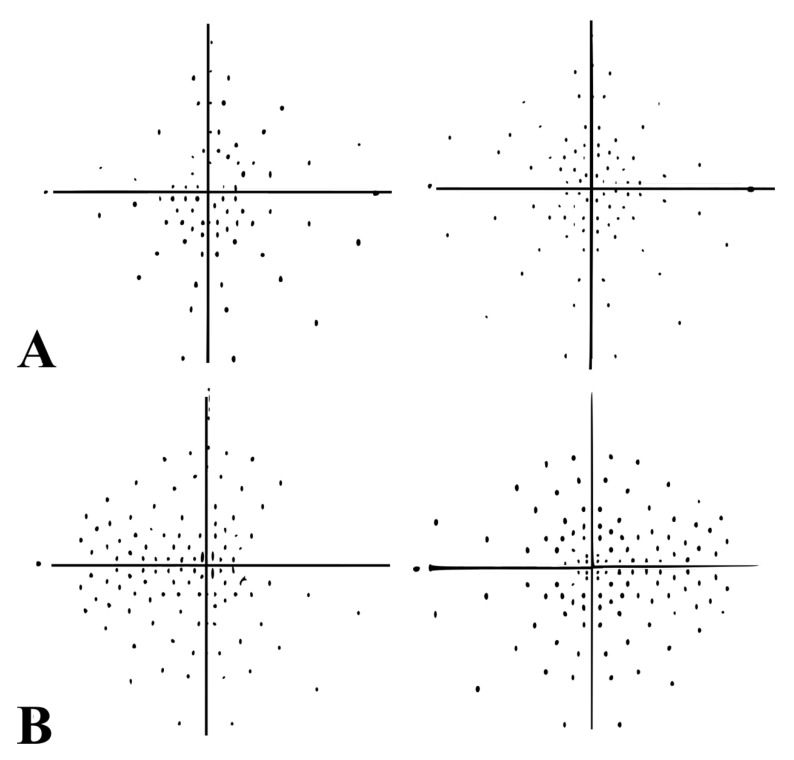
Humphrey Visual Field Analyzer, Central 30 for both the eyes. (**A**) Initial visual field examination of the right eye (left side) shows a macular-sparing with inferior hemifield and temporal-superior quadrant scotoma. Visual field in the left eye (right side) was completely normal. (**B**) After one year of follow-up, the right eye (left side) showed persistent severe loss in visual field examination. Humphrey automated perimetry revealed severe widespread visual field loss with central-sparing in the left eye (right side).

**Figure 2 diagnostics-11-02325-f002:**
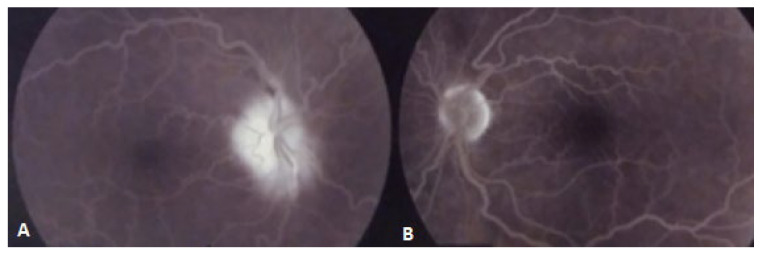
(**A**) Late-phase fundus fluorescein angiogram of the RE showed blurred margins of the optic disc with staining. (**B**) Left optic nerve head was normal. Both eyes showed vascular tortuosity.

**Figure 3 diagnostics-11-02325-f003:**
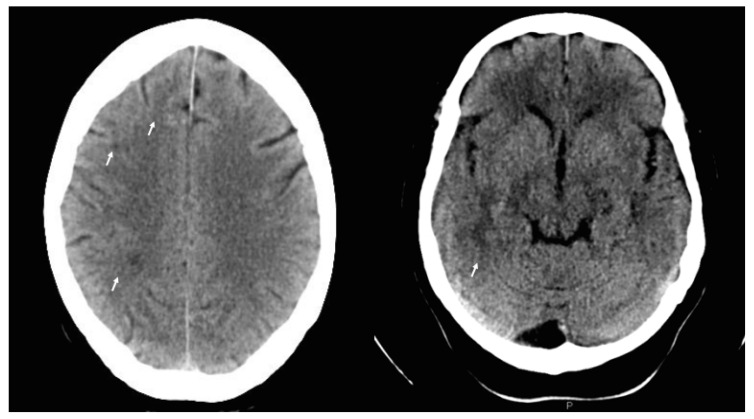
Cranial computed tomography (CT) scan showed small diffuse areas of white matter attenuation (leukoaraiosis; indicated with arrows and appearing as a low signal on CT).

**Figure 4 diagnostics-11-02325-f004:**
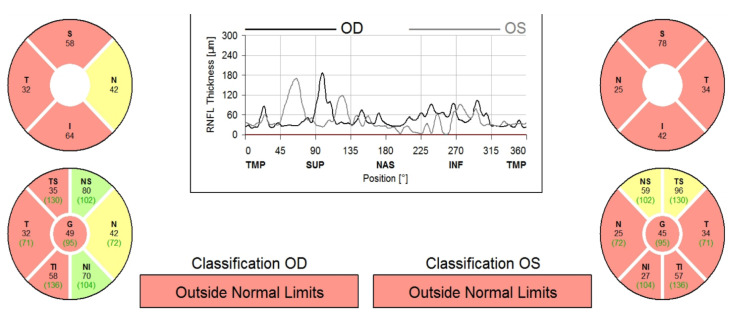
SD-OCT thickness map showing important loss of retinal nerve fiber layer (RNFL) in both eyes (red) following NA-AION, with perseverance of the right eye nasal quadrant (green) and relative perseverance of the left eye superior quadrant (yellow). S (superior), T (temporal), N (nasal), I (inferior), TS (Superior temporal), NS (superior nasal), TI (inferior temporal), NI (inferior temporal), G (global), OD (right eye), OS (left eye), TMP (temporal), SUP (superior), NAS (nasal), INF (inferior).

## Data Availability

All datasets on which the conclusion of the paper relies are available to readers in PubMed, with references. Patient’s data and medical tests are available in clinic history.

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
