# Peer review of "Non-Simultaneous Bilateral Ischemic Optic Neuropathy Related to High Altitude and Airplane Flight in a Patient with Cerebral Small Vessel Disease"

_diagnostics, 2021, doi:10.3390/diagnostics11122325_

Round 1
Reviewer 1 Report
Boned-Murillo et al. wrote a very interesting case report describing the “Non simultaneous bilateral ischemic optic neuropathy related to high altitude and airplane flight in a patient with cerebral small vessel disease”. The manuscript represents an interesting way to review the newest scenarios for ocular degenerations. I suggest only minor revisions needed to update and improve the reliability of the paper:
• The manuscript, even if it is a case report, is too short, and should be better detailed.
• The authors should better update the literature about the role of vasculogenesis and optical neuropathy, well known causes of ocular pathologies. Regarding these, I suggest to add the following references to the manuscript PMID: 33233546 and PMID: 34440511.
• Finally, manuscript requires English revisions and typos correction.
Author Response
Review 1.
Dear reviewer
Thank you for the comprehensive review and valuable comments on our manuscript entitled ‘’Non simultaneous bilateral ischemic optic neuropathy related to high altitude and airplane flight in a patient with cerebral small vessel disease‘’. We have revised the paper in accordance with the reviewer’s suggestions. In the following pages we will address each comment one by one. We hope our paper will be acceptable to be published in ‘’Diagnostics’’.
Point 1. The manuscript, even if it is a case report, is too short, and should be better detailed.
Manuscript was revised and some parts were rewriten, with a better detailed introduction (Page 1, 1st paragraph, line 23-34), discussion and figure legends clarification.
Point 2. The authors should better update the literature about the role of vasculogenesis and optical neuropathy, well known causes of ocular pathologies. Regarding these, I suggest to add the following references to the manuscript PMID: 33233546 and PMID: 34440511.
References 11 (Scimone, C; Alibrandi, S; Scalinci, SZ et al. Expression of Pro-Angiogenic Markers Is Enhanced by Blue Light in Human RPE Cells. Antioxidants (Basel). 2020. 20;9(11):1154) and 12 (Rinaldi, C; Donato, L; Alibrandi, S et al. Oxidative Stress and the Neurovascular Unit. Life (Basel). 2021;11(8):767) were added and the following setences were incorporated in the text: ‘’ Systemic, endocrine, neural factors, and oxidative stress disorders of the neurovascular unit (NVU) can contribute to the development of a brain injury and plays a major role in dysregulation, neuroinflammation, and neurodegeneration after a stroke. Complex communication and cell-to-cell communication is essential 11, 12, for example pericytes modulators such as the NADPH oxidase Nox4 (Reactive oxygen species (ROS) ) are significantly increased under hypoxic conditions increasing MMP-9 and may increase the size of the infarct area by playing a harmful role in the acute phase of ischemic stroke ‘’ (Page 4, 3rd paragraph, line 118-120).
Point 3. Finally, manuscript requires English revisions and typos correction.
English revisions proof reading was performed as suggested.
Reviewer 2 Report
The manuscript contains a case report of a woman who had optic neuropathy that developed in each eye related to separate exposures to high altitude (once during an airplane flight and once during a high-altitude drive). This represents a rare pathology, and the authors discuss that small vessel disease in this patient may be part of the pathology in this individual. This case is interesting in that the patient developed optic neuropathy in both eyes during separate instances of high altitude exposure. The article would benefit from some clarifications, and some corrections to figures. The article as written is directed to an ophthalmologist audience, so the authors may want to consider adding a little more background to orient the audience of Diagnostics.
- The authors may want to broaden the introduction some, to help those readers who are not ophthalmologists understand better what NA-AION is.
- Figure 1. Likewise, the authors may wish to give more background to orient readers to the meaning of the optic field diagrams. In this figure as well, the left vs right eyes are not well labeled, and the order between Fig 1A and 1B is reversed (ie in Fig 1A the right eye is on the left and the left eye is on the right, while this order is reversed in Fig 1B)
- Figure 2: the legend refers to a part A and part B, but these are not labeled in the figure.
- Figure 3: the legend refers to arrowheads that are not present in the figure.
- Figure 4: again, it would be helpful for non-ophthalmologist readers if the authors gave more explanation of what OCT reveals about retinal injury related to optic neuropathy, and some orientation to what the diagrams mean.
- Discussion, lines 106-108: although small vessel disease was also present in this patient, how likely is it that her NA-AION is related to this? If there is more information in the literature about a connection between SVD and NA-AION I would suggest adding more of a discussion here. Otherwise, the statement in line 106 that “NA-AION seems to be the result from SVD that supply the ONH” would seem to be over-interpreting the data from only a single patient who happened to have both optic neuropathy and SVD.
Author Response
Review 2.
Dear reviewer
Thank you for the comprehensive review and valuable comments on our manuscript entitled ‘’ Non simultaneous bilateral ischemic optic neuropathy related to high altitude and airplane flight in a patient with cerebral small vessel disease ‘’. We have revised the paper in accordance with the reviewer’s suggestions. In the following pages we will address each comment one by one. We hope our paper will be acceptable to be published in ‘’Diagnostics’’.
- Point 1. The authors may want to broaden the introduction some, to help those readers who are not ophthalmologists understand better what NA-AION is.
As suggested by reviewers, introduction was clarified as follows: ‘’ with a reported annual incidence of 2.3-10.2 per 100,000 population over 50 years, with significantly higher frequency in Caucasians 1. Lower visual field is usually affected with a normal visual acuity in 40% of patients, relative afferent pupillary defect (RAPD) is always present and disc oedema is observed sometimes associated to flame shaped haemorrhages at the margins 1. Contrary to the arteritic form (A-AION), known as Giant Cell Arteritis (GCA), it is not necessarily associated with the elderly, a constitutional symptoms or elevated level of erythrocyte sedimentation rate (ESR), C-reactive protein (CRP) and thrombocytosis. Vision can worsen over the 2 week period following initial presentation and typically stabilizes by 2 months, visual acuity improvement occurs in less than 10% of patients as visual field improvement and there is never complete recovery of both acuity and visual field 1. Recurrence in the affected eye from 3% to 8% or fellow eye involvement from 15% to 24% over 5 years 1. ‘’ (Page 1, 1st paragraph, line 23)
- Point 2. Figure 1. Likewise, the authors may wish to give more background to orient readers to the meaning of the optic field diagrams. In this figure as well, the left vs right eyes are not well labeled, and the order between Fig 1A and 1B is reversed (ie in Fig 1A the right eye is on the left and the left eye is on the right, while this order is reversed in Fig 1B)
Optic field diagram was explained in the figure legend as follows: ‘’ Figure 1. Humphrey Visual Field Analyzer, Central 30 for both the eyes. A. Initial visual field examination of the right eye (left side) shows a macular-sparing with inferior hemifield and temporal-superior quadrant scotoma. Visual field in the left eye (right side) was completely normal. B. After one year of follow-up, the right eye (left side) showed persistent severe loss in visual field examination. Humphrey automated perimetry revealed severe widespread visual field loss with central-sparing in the left eye (right side) ‘’ (Page 2, line 61-65).
In the same way, Figure 1 and figure legends refered to optic field diagrams were modified according to suggestions (Page 2, 2nd paragraph, line 61).
- Point 3. Figure 2: the legend refers to a part A and part B, but these are not labeled in the figure.
Figure 2 part A and part B were correctly labeled (Page 3, line 68).
- Point 4. Figure 3: the legend refers to arrowheads that are not present in the figure.
The arrowheads in Figure 3 have been inserted (Page 3, line 72) and the figure legend was modified as follows: ‘’Cranial computed tomography (CT) scan showed small diffuse areas of white matter attenuation (leukoaraiosis; indicated with arrows and appearing as a low signal on CT’’ (Page 3, line 71-72).
- Point 5. Figure 4: again, it would be helpful for non-ophthalmologist readers if the authors gave more explanation of what OCT reveals about retinal injury related to optic neuropathy, and some orientation to what the diagrams mean.
Figure legend corresponding to Figure 4 was described with the following sentence ‘’SD-OCT thickness map showing important loss of retinal nerve fiber layer (RNFL) in both eyes (red) following NA-AION, with perseverance of the right eye nasal quadrant (green) and relative perseverance of left eye superior quadrant (yellow). S (superior), T (temporal), N (nasal), I (inferior), TS (Superior temporal), NS (superior nasal), TI (inferior temporal), NI (inferior temporal), G (global), OD (right eye), OS (left eye).’’ (Page 4, 1st paragraph, line 86-90)
- Point 6. Discussion, lines 106-108: although small vessel disease was also present in this patient, how likely is it that her NA-AION is related to this? If there is more information in the literature about a connection between SVD and NA-AION I would suggest adding more of a discussion here. Otherwise, the statement in line 106 that “NA-AION seems to be the result from SVD that supply the ONH” would seem to be over-interpreting the data from only a single patient who happened to have both optic neuropathy and SVD.
To improve the discussion, the following reference was added (Kim, MS; Jeong, HY; Cho, KH et al. Nonarteritic anterior ischemic optic neuropathy is associated with cerebral small vessel disease. PLoS One. 2019. 14; 14(11):e0225322) and the following sentence was described: ‘’ Association between cerebral SVD and NAION was studied by Kim et al 15 who found an association after adjusting for age, sex, and medical histories ‘’ (Page 5, 1st paragraph, line 133-135).
Reviewer 3 Report
The Manuscript-Case Report “Non simultaneous bilateral ischemic optic neuropathy related to high altitude and airplane flight in a patient with cerebral small vessel disease”, co-authored by Ana Boned-Murillo et al. represent a rare case of non-simultaneous bilateral acute presentation of non-arteritic anterior ischemic optic neuropathy (NA-AION) related to high altitude and airplane flight in a patient with small vessel disease. Although NA-AION is considered the most frequent type of acute optic neuropathy, the clinical history of this patient is particularly interesting, and accurately informs about this rare case in which NA-AION occurs asynchronously but related with a identical situation.
The work is well written and easy to read. The medical history is very well detailed, and the discussion of the results raises a plausible etiopathogenesis.
The manuscript should, however, be modified in some respects to improve quality.
1.- Improve the Quality of Figure 1, eliminating the background to better perceive the points and the text, which appears out of focus;
2.- The arrowheads in figure 3 have disappeared
3.- It is necessary to review the edition since some sub-indices (e.g. CO2 instead of CO2) and the location of the reference numbers are displaced.
Author Response
Review 3.
Dear reviewer
Thank you for the comprehensive review and valuable comments on our manuscript entitled ‘’ Non simultaneous bilateral ischemic optic neuropathy related to high altitude and airplane flight in a patient with cerebral small vessel disease ‘’. We have revised the paper in accordance with the reviewer’s suggestions. In the following pages we will address each comment one by one. We hope our paper will be acceptable to be published in ‘’Diagnostics’’.
Point 1.- Improve the Quality of Figure 1, eliminating the background to better perceive the points and the text, which appears out of focus.
Figure 1 was modified has suggested to improve visualization and clarify information.
Point 2.- The arrowheads in figure 3 have disappeared
The arrowheads in Figure 3 have been inserted (page 3, line 70).
Point 3.- It is necessary to review the edition since some sub-indices (e.g. CO2 instead of CO2) and the location of the reference numbers are displaced.
As suggested, sub-indices of the word CO2 was corrected to CO2, instead of CO2 (Page 4, 3th paragraph, line 113).
Location of the reference numbers has been replaced as well and a new reference was added to the manuscript (reference number 10: De Bats, F; Gambrelle, J; Feldman, A et al. Survenue d’une neuropathie optique ischémique antérieure aiguë bilatérale en haute altitude: rôle de l’intolérance à l’hypoxie. Journal Français d'Ophtalmologie. 2010; 33(10): 724-727) as follows: ‘’The first one corresponded with a unilateral NA-AION in a man who stayed at an altitude of 5472 meters above sea level for three months9. Moreover, it was reported a bilateral NA-AION case of a 66 years old woman during a trek at altitude higher than 2500 m10’’ (Page 4, 3th paragraph, line 104-108) or ‘’…As Moran et al. 13 concerned…’’ (Page 4, 3th paragraph, line 131).